# Mesogeos: A multi-purpose dataset for data-driven wildfire modeling in the Mediterranean

**Spyros Kondylatos**[1,2]**, Ioannis Prapas**[1,2]**, Gustau Camps-Valls**[2]**, and Ioannis Papoutsis**[1]

[1]Orion Lab, Institute for Astronomy, Astrophysics, Space Applications, and Remote Sensing,
National Observatory of Athens
[2]Image Processing Laboratory (IPL), Universitat de València
{skondylatos, iprapas, ipapoutsis}@noa.gr
{gustau.camps}@uv.es

## Abstract

We introduce Mesogeos[1], a large-scale multi-purpose dataset for wildfire modeling in the Mediterranean. Mesogeos integrates variables representing wildfire drivers (meteorology, vegetation, human activity) and historical records of wildfire ignitions and burned areas for 17 years (2006-2022). It is designed as a cloud-friendly spatio-temporal dataset, namely a datacube, harmonizing all variables in a grid of 1km x 1km x 1-day resolution. The datacube structure offers opportunities to assess machine learning (ML) usage in various wildfire modeling tasks. We extract two ML-ready datasets that establish distinct tracks to demonstrate this potential: (1) short-term wildfire danger forecasting and (2) final burned area estimation given the point of ignition. We define appropriate metrics and baselines to evaluate the performance of models in each track. By publishing the datacube, along with the code to create the ML datasets and models, we encourage the community to foster the implementation of additional tracks for mitigating the increasing threat of wildfires in the Mediterranean.

## 1 Introduction

Wildfires play a key role in the ecosystem [1–4], yet they present risks to both humans and the environment [5]. The threat is inflated by climate change, which aggravates the frequency and extremity of wildfire events [6], particularly in Mediterranean-type climate regions [7, 8]. The changes are expected to be more and more prevalent in the following years [9]; thus, there is a pressing need for innovative solutions to enhance wildfire preparedness and management, enabling adaptation to evolving conditions. The development of such solutions is hampered by the complexity to model wildfires, resulting from the dynamic interactions between several fire drivers such as climate, vegetation, and human activity [10], operating across different spatial and temporal scales.

Traditional models [11–13] ignore these intricate interactions. In contrast, Machine Learning (ML) offers the potential to capture them in a data-centric manner. Nevertheless, the application of ML in the context of wildfires requires careful consideration [14]. The wildfire occurrence is stochastic, which means that the same environmental conditions may lead or not to a fire ignition. Moreover, wildfires are rare events that can lead to imbalanced or sparse datasets. Despite the challenges, ML has been employed successfully in several applications [15]. Particularly, Deep Learning (DL) has been suggested as a method for modeling Earth System problems, including wildfires [16, 17].

---

[1]Inspired from the Greek word M$\varepsilon\sigma\acute{o}\gamma\varepsilon\iota o\varsigma$ (Mesógeios), widely adopted to refer to the Mediterranean Sea.

37th Conference on Neural Information Processing Systems (NeurIPS 2023) Track on Datasets and Benchmarks.

Although the potential of DL in wildfire modeling appears promising, its adoption is still not widespread. One major obstacle is the limited availability of extensive datasets necessary to support its utilization. The vast amount of data required to model wildfires at a larger scale presents difficulties in the collection and curation of the data. The data sources are often scattered across different platforms and become available in diverse formats and resolutions. Thus, the community lacks a large-scale dataset suitable for various ML tasks in the context of wildfires.

In this work, we introduce *Mesogeos*, an extensive multi-purpose dataset designed to support the development of ML models for various wildfire applications in the Mediterranean. It contains a complete set of variables associated with fire drivers, i.e. meteorological conditions, vegetation characteristics, and anthropogenic factors. It also encompasses past burned areas, fire ignition points, and burned area sizes that can serve as predictands for diverse ML tasks. Mesogeos is harmonized in a standard spatiotemporal grid format, namely a datacube [18], with a daily temporal resolution and a spatial resolution of $1km \times 1km$, containing data from 2006 to 2022. The datacube structure facilitates the extraction of ML-ready datasets for numerous applications. To the best of our knowledge, Mesogeos is the largest harmonized, multi-purpose dataset for data-driven wildfire modeling.

To demonstrate the datacube's potential applications, we extract from it two ready-to-consume ML datasets: one tailored for the next day's wildfire danger forecasting and one for burned area size prediction, given the ignition. We employ DL models to establish benchmarks for the two datasets. Furthermore, we propose several additional directions for utilizing the dataset, suggesting its capabilities for addressing other wildfire-related applications. To encourage further research and facilitate the development of similar datasets, we openly publish the Mesogeos datacube, the derived datasets and models, and the code used to generate them [19]. We also provide a github repository: https://github.com/Orion-AI-Lab/mesogeos and a website for the project: https://orion-ai-lab.github.io/mesogeos/ with information on how to use the data and code. These resources can be a valuable reference for future implementations and extractions of similar datasets.

## 2 Related Work

DL has demonstrated successful applications in various tasks related to wildfires. Huot et al. [20] have built segmentation models for predicting fire danger with U-Net-type architectures. Radke et al. [21] developed FireCast, a fire spread model leveraging Convolutional Neural Networks (CNNs) that demonstrated superior performance compared to physics-based models. Similarly, Hodges and Lattimer [22] and Burge et al. [23] employed DL techniques to predict fire evolution by training on fire simulations. Lastly, Ba et al. [24] addressed the fire detection task, by developing SmokeNet, a CNN-based model that was trained to predict hotspots, as provided by the Moderate Resolution Imaging Spectroradiometer (MODIS) Active Fire (AF) data product [25]. Although these studies showcased the potential of DL in various wildfire applications, the datasets used in each work remain unpublished, thus it is impossible for the community to reproduce or improve the results.

When it comes to modeling wildfires, many studies rely on satellite-derived data, such as AF products that detect thermal anomalies or burned area products that locate rapid reflectance changes. MODIS and VIIRS satellites offer such openly accessible products and are commonly used due to their high temporal resolution, offering daily global coverage. The MODIS AF product exhibits a spatial resolution of $1km \times 1km$ and has been generating data since 2002. It operates by employing thermal sensors to identify anomalous thermal signatures associated with ongoing fires [25]. The VIIRS satellite, introduced in 2012, follows a similar AF detection methodology for fires, holding an enhanced spatial resolution of $375m \times 375m$, which leads to a better response to relatively small fires and possesses an improved nighttime performance [26, 27]. Several studies have been undertaken to assess the quality of these products by comparing their outcomes against human-collected fire databases. These analyses have brought to light certain limitations associated with their utilization for the assessment of wildfires. In the United States and China, MODIS demonstrated a moderate level of concurrence with actual fire data [28, 29]. Moreover, both MODIS and VIIRS products exhibited disagreements when evaluated against real fire occurrences in Turkey, a Mediterranean-type region, with more favorable results observed for larger fires [30].

Alternative datasets sourced from MODIS include MOD14A1 [31] and MCD64A1 [32]. The former is an open-source product, containing a collection of daily fire mask composites at a spatial resolution

of $1km \times 1km$. The latter, becomes available at a spatial resolution of $500m \times 500m$, mapping the spatial extent and approximate date of biomass burning worldwide. Several validation studies have been undertaken to assess the accuracy of these datasets, revealing instances of disagreement between their outputs and reliable fire records [33–36]. Apart from MODIS products, there are several other publicly available fire datasets derived from Earth Observation satellites. These include global datasets such as FRY [37] and Fire Atlas [38] which are designated to deliver the total burned area of fires and GlobFire [39] which provides daily fire perimeters. In Europe, the European Forest Fire Information System (EFFIS) [40] provides accurate burned area estimates following a semi-supervised approach that uses different satellite sensors, estimating about $95\%$ of the total area that burns in Europe every year [41]. The EFFIS burned area product is used in this work because of its improved accuracy in the Mediterranean region.

While these datasets provide only fire data, a comprehensive wildfire analysis and modeling needs to incorporate variables related to fire drivers, such as vegetation, weather, drought, and topography. In this direction, several studies have published datasets that integrate fire targets with variables related to fire ignition and spread. These datasets often have limitations, such as focusing on specific small-scale regions or exhibiting coarse spatial and temporal resolutions. Furthermore, they are typically designed for specific tasks tailored to a single ML objective. Kondylatos et al. [42] have published a dataset covering Greece, which is specifically designed for forecasting the next day's wildfire danger. Though they also publish a datacube, its applications are limited by its small size, only covering a part of the eastern Mediterranean. Huot et al. [43], Singla et al. [44], Diao et al. [45] have introduced datasets designated for wildfire spread prediction in the continental US. The former relies on the MODIS AF product as the target variable, while the others utilize the VIIRS AF product. Moreover, Sayad et al. [46] have shared a dataset tailored to wildfire modeling in a small region of Canada, recording a limited number of fire events. Finally, Prapas et al. [47] presented a global dataset for seasonal fire danger forecasting, but in a coarse spatial and temporal resolution.

**Comparison with Existing Datasets and Contributions.** Mesogeos is a large-scale, versatile, multi-purpose dataset designed to cater to a multitude of ML tasks related to wildfire modeling. It sets itself apart from datasets [42–46], which focus solely on single ML tasks. It offers a broad scope, by encompassing a wide range of daily inputs covering many relevant fire drivers across the entire area of interest. It is provided in a cloud-optimized datacube structure, offering spatio-temporal metadata, that uniquely associates data points with specific date, longitude, and latitude values. This structure empowers researchers to select subsets in any dimension, retrieve variables, calculate new ones, or even augment the datacube with other data. Such inherent flexibility simplifies data access, facilitates the extraction of diverse ML datasets, and enables the expansion of the dataset. Thus, it allows its adaptation to a wide range of ML applications, based on individual research needs. Moreover, by leveraging the EFFIS burned area product while refining the provided ignition dates of fires through cross-comparison with the MODIS AF product (as illustrated in Section 3), we achieve a more reliable representation of burned areas and fire ignitions compared to existing datasets [25, 26, 31, 32, 37–39, 48]. Finally, it is worth noting that Mesogeos is the first dataset of this resolution and scale tailored for wildfire modeling in the Mediterranean region, a fire-prone area that has lacked dedicated datasets of this nature.

## 3 Mesogeos Datacube

**General information.** The Mesogeos dataset is structured as a spatio-temporal datacube with three dimensions: longitude, latitude, and time. The datacube encompasses 27 variables related to meteorology, vegetation, land cover, and human activity. All these data are well-known fire drivers and can be used as predictors in wildfire-related applications. Mesogeos also includes historical burned areas, ignitions, and burned area sizes as separate variables. It has $1km \times 1km \times daily$ resolution and contains the values of the variables covering the period from 2006 to 2022. It incorporates data from the wide Mediterranean area and spans a total area of $4714km \times 1753km$ and 6026 days.

**Data Sources.** Hantson et al. [49] study the complex interactions between the variables that control fire. They divide fire controllers into three main categories: human, weather, and vegetation. Weather conditions and vegetation play a crucial role in determining the rates of fuel drying and therefore affecting fire occurrence and spread. Topography also influences fire behavior as fire fronts travel faster uphill because of the upward convection of heat. While natural factors such as weather,

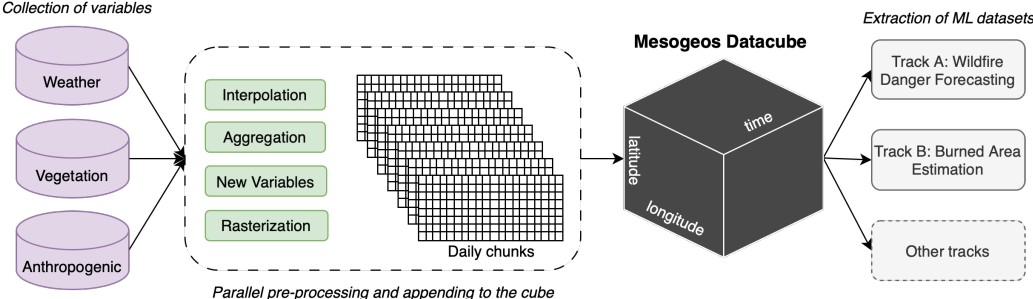

Figure 1: Pipeline of the datacube construction. The data are collected from various sources. The data inputs are pre-processed using interpolation, aggregation, calculation of new variables, and rasterization and the final daily chunks are appended in the datacube on the corresponding date. This process runs in parallel for multiple days to enhance the efficiency of the process. After the creation of the datacube, the ML datasets are extracted from it.

vegetation, and fuel load impact fire occurrence, human activity-related outcomes such as intentional or accidental fire ignition, land conversion, and population density also significantly shape fire regimes. In this study, in an attempt to cover all the factors influencing fire occurrence, we collect data sources containing information about all the aforementioned fire drivers.

The meteorological data (temperature, wind speed, wind direction, dewpoint temperature, surface pressure, relative humidity, total precipitation, surface solar radiation downwards) are collected from the ERA5-Land database [50], which contains historical hourly land weather measurements from 1950 to today. We use day's and night's land surface temperature [51], Normalized Difference Vegetation Index (NDVI) [52], and Leaf Area Index (LAI) [53] from MODIS and soil moisture index from the European Drought Observatory (EDO) [54]. These data are used as proxies for the vegetation status and drought. Distance from roads and population are downloaded from Worldpop [55] and are used as indicators of human activity. At the same time, topography data, i.e. elevation, slope, aspect, and curvature are gathered from the Copernicus DEM - Global Digital Elevation Model (COP-DEM) [56]. The land cover classes are collected from the Copernicus Climate Change Service [57]. The burned areas come from EFFIS. Finally, MODIS AF product [25] is used to estimate ignition cells and the ignition date.

**Datacube Creation.** Creating a unified dataset that stores all wildfire-related information in a standard format will later permit an easy extraction of ML-ready datasets for different tasks. Consequently, we have opted to gather and harmonize all the data into a spatio-temporal datacube format to leverage this structure's various capabilities regarding easy access, manipulation, and extraction of data. Nevertheless, the creation of such a datacube poses significant challenges. The substantial volume of data (TBs of unprocessed data) necessitates significant downloading and storage capabilities. Additionally, the data are sourced from different providers, each with their own access points and formats (e.g. vector or raster), making data acquisition a challenging task. Furthermore, the original resolution of each variable varies, requiring harmonization to match the expected resolution of the datacube. Consequently, the construction of the datacube demands careful and efficient development for minimizing time and resource requirements.

To address the challenges above, we create the pipeline illustrated in Figure 1 for the creation of the datacube. We follow these steps: Firstly, we collect and store all relevant variables from the various input sources. Subsequently, we construct the datacube's structure by generating a grid of dimensions $1km \times 1km \times 1-day$ and defining daily chunks. The daily chunking applies independently to each variable, which means that values are stored in different files for each day. Finally, we append each input source to the datacube on a day-to-day basis to prevent memory errors. For each day, we perform all the necessary pre-processing steps, such as converting data into raster format, conducting temporal or spatial interpolation/aggregation, changing coordinate systems, and doing variable calculations. Then, we store the values in the chunk that refers to the corresponding date. As chunks are stored independently, we make this process totally in parallel. We use the xarray [58] python package for development. Using the default Zarr [59] compression, the datacube occupies a storage space of 648 GB, while the memory needed to load the datacube is much larger, at around 3.2 TB, assuming 32-bit

floats for the dynamic variables. The code for creating this datacube is made available and can be consulted to further enhance the existing dataset. Notably, this pipeline can be adapted with minor adjustments for generating similar datacubes applicable to various Earth science domains. For a more comprehensive understanding of the pre-processing procedures undertaken for each variable, please refer to the Supplementary Material.

**Burned Areas Dataset.**   The burned areas dataset provided by EFFIS is an improvement over the MODIS products, offering a more reliable and credible resource. EFFIS enhances the burned area data obtained from MODIS by employing a semi-supervised processing of imagery from various satellites, i.e. Sentinel-2, and VIIRS. This process involves semi-automatic procedures aimed at enhancing the quality of fire maps [41]. Despite its improved quality, it is important to note that the dataset does not offer the same coverage of burned areas across all Mediterranean countries throughout the specified timeframe, resulting in the absence of data for certain countries in specific years. This is further analyzed in the Supplementary Material.

**Ignition Date Calculation.**   The start dates of the fires provided by EFFIS may not always correspond to the date of ignition of the fire [41]. However, the accurate calculation of the first detection of a wildfire is extremely important in order to avoid data leakage and enhance the transparency and precision of the training process. For this, we implement a method that involves intersecting burned areas from EFFIS with AF obtained from MODIS. We use each burned area identified by EFFIS as a representative instance of a distinct fire event. We then select hotspots from a $1km$ spatial buffer zone surrounding the burned area and a temporal buffer of 7 days around the date of the ignition as provided by EFFIS. From this selection, we identify the hotspot with the oldest date within the buffer as the ignition point of the fire and its date as the ignition date of the fire. We discard the specific wildfire from the dataset if no hotspots are detected within the designated buffer zone. Notably, MODIS AF are used solely for ignition date refinement and ignition point identification and are not employed as primary anchors for fire events. Despite the improvements achieved through our approach, it is essential to recognize that some misalignments in the ignition dates may persist.

## 4   Machine Learning tracks

### 4.1   Track A: Wildfire danger forecasting

**Task formulation.**   For a given cell and a given day $t$, we define fire danger as the probability of a fire occurring on the day $t$ and becoming large, given the values of the different fire drivers $x_t$ in the preceding days. We assume that a wildfire exceeding 30 hectares indicates high wildfire danger. To measure this danger, we treat the ignition point of the fire as a representative point of the event and the final burned area size resulting from it as an indicator of the corresponding danger level. Conversely, low wildfire danger is associated with the absence of any wildfire within a specified buffer zone surrounding a given pixel. In alignment with prior research in the field of data-driven wildfire danger forecasting [20, 42, 60], the task is formulated as a binary ML classification problem. One class signifies increased danger, while the other represents low-danger instances. However, in this work, we slightly modify the standard classification loss, involving a weighting scheme that considers the burned area size of each distinct event. This modification aims to interpret a more significant fire expansion as an indication of higher danger. The resulting softmax probabilities of the trained classification model serve as indicators of the level of fire danger.

**Dataset extraction.**   We extract a time-series dataset, consisting of days $t-1, t-2, ..., t-30$ of the dynamic input observations and the static features repeated in time. Positive class examples consist of the ignition points of the fires that started on the day $t$. For the negative class examples, we select cells outside a buffer of $62km$ from any fire that started on this day to mitigate the risk of choosing cells that imply great danger but did not burn. Moreover, we follow the sampling strategy as in [42] and sample i) two times more negatives than positives, ii) the negatives following the land cover distribution of the positives.

**Experimental Setup.**   For the experiments, we use a Long Short-Term Memory (LSTM) architecture [61] and the encoder of a Transformer model [62] as standard models for time-series data. Moreover, we employ a Gated Transformer Network (GTN) [63] that uses the attention mechanism both in time

and in variables, which could aid in modeling the complex interactions of the variables in the current task. The models are optimized using the cross-entropy (CE) loss. To let the ML models learn to assign greater danger to larger fires, we weigh the loss based on the size of the wildfire's burned area. For this, we multiply the standard CE loss value of a given sample by the corresponding burned area size resulting from it. In practice, in order to prevent the larger fires from totally dominating the learning process, we apply a logarithmic transformation to the burned area size multiplier, in an attempt to narrow the penalization gap between small and large fires. Negative samples are assigned weights equal to the minimum burned area size among the positive samples. This ensures that the negatives, representing low-danger instances, receive adequate attention during the training process, but not more than any high-danger instance. The hyperparameters for each model are tuned separately using the validation set. The years $2006 - 2019$ are used for the training set, $2020$ is used as the validation set, and the years $2021 - 2022$ are used as a test set. The final dataset consists of $25722$ samples ($8574$ positives and $17148$ negatives), from which $19353$ ($6451$ positives and $12902$ negatives) are in the training set, $2262$ ($754$ positives and $1508$ negatives) are in the validation set and $4107$ ($1369$ positives and $2738$ negatives) are in the test set. All the available input variables from the datacube are used in the experiments. They are all normalized before passing into the model. Precision, Recall, and Area Under Precision-Recall Curve (AUPRC) are used as metrics for the evaluation of the performance of the models. The details about the architectures of the models and the hyperparameters are provided in the Supplementary Material.

Table 1: Results of the fire danger forecasting track

| Model | Precision | Recall | $F1$ | AUPRC |
|---|---|---|---|---|
| LSTM [61] | 0,763 | **0,812** | **0,786** | 0,853 |
| Transformer[62] | **0,802** | 0,759 | 0,780 | 0,856 |
| GTN [63] | 0,781 | 0,790 | **0,786** | **0,858** |

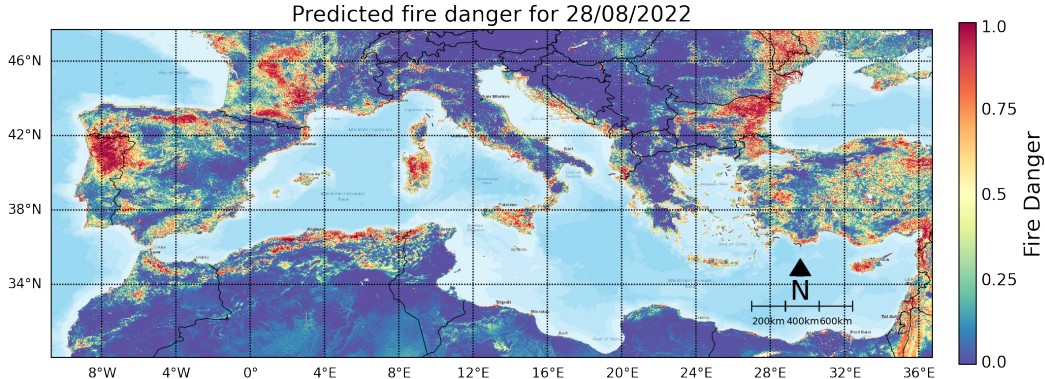

Figure 2: A wildfire danger map of the Mediterranean, produced by the Transformer model. The fire danger is indicated by the softmax probabilities of the trained model.

**Results.**    The results of the models are shown in Table 1. The promising results from all the models show that they can distinguish between high and low fire danger instances. It should be noted that the optimal performance model varies for each metric, thus making it difficult to define the overall best-performing model. In addition to the quantitative metrics, we also provide a visualization map generated by the Transformer model, presented in Figure 2. The map displays the model's softmax probabilities, indicating fire danger levels. This provides an example of how a daily fire danger map can be generated for operational scenarios in the Mediterranean region. The high spatial resolution of the dataset employed in this study could enable improved fire management strategies and demonstrate potential operational benefits. In this example, concerning fire danger for a day in the summer of 2022 notable variations in fire danger are observed across different areas of the Mediterranean, as well as within individual regions of each country.

## 4.2 Track B: Final Burned Area Prediction

**Task formulation.** This track focuses on predicting the likely extent of a wildfire's final burned area, given the ignition point and a set of variables available at the time of ignition, representing the fire drivers inside a neighborhood around the ignition point. These fire drivers encompass factors that influence fire behavior and spread. Thus, the objective of the ML task is to estimate the likelihood of the neighboring pixels surrounding the ignition point, to be eventually contained within the final burned area of the wildfire. The final burned area prediction is treated as a segmentation task, with two classes, indicating whether a pixel will experience burning or remain unaffected by the ignited fire. The resulting softmax probabilities of the trained model serve as indicators of the likelihood of a pixel being burnt.

**Dataset extraction.** For every fire event, we extract $64km \times 64km$ patches, that are centered around the fire's ignition point, usually containing the whole burned area of a given fire event. The extracted samples include all the values of the variables of the datacube for the date of the fire's occurrence.

**Experimental Setup.** The $64 \times 64$ patches are randomly cropped to $32 \times 32$ during the training process. This approach ensures that the ignition point remains within the patch while preventing the model from generating a bias towards fire expansion solely from the central cell. As this is a segmentation task, we use the U-Net architecture [64] with an EfficientNet-B1 [65] encoder. Different input variables are stacked as separate channels. The cross-entropy loss is used to train the models' parameters. To define a baseline for the task, we train an additional model that uses as input only the ignition points. We do a temporal split to avoid leaking data from fire events happening close in time, using $2006 - 2019$ for training (12550 samples), 2020 for validation (1781 samples), and $2021 - 2022$ for testing (3527 samples). The loss in the validation is used for early stopping. Input variables are scaled with the minimum, and maximum values in the range $[0, 1]$ before being served as inputs to the model. As evaluation metrics, we report the CE loss and the AUPRC. The exact architecture and the values of the hyperparameters are provided in the Supplementary Material.

Table 2: Results of the final burned area prediction track

| Model | CE Loss | AUPRC |
|---|---|---|
| U-Net (only ignitions) | 0.0177 | 0.394 |
| U-Net (all variables) | 0.0166 | 0.418 |

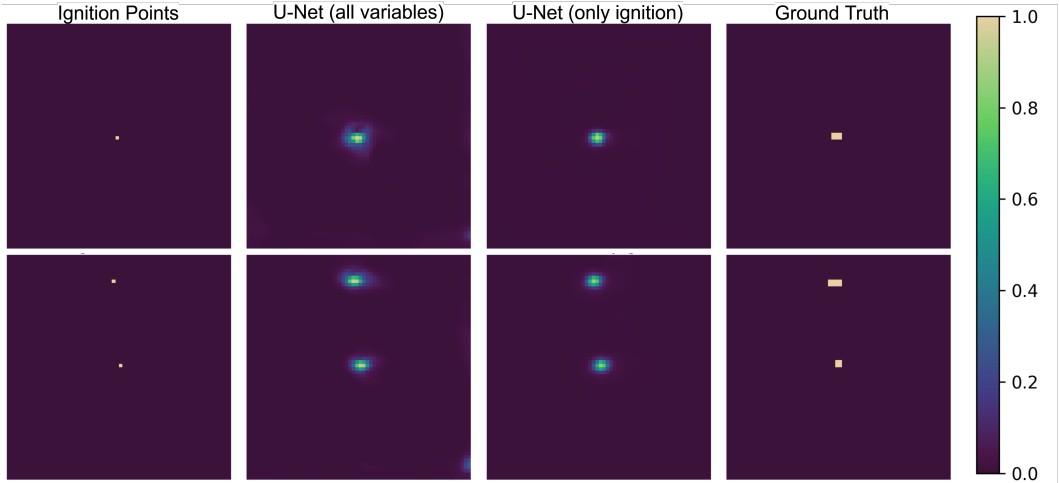

Figure 3: Two examples of predictions (softmax values for the positive class) from the U-Net using *all variables*, and the U-Net using *only ignition* points. The predictions are presented together with binary maps representing the initial *ignition points* and the *ground truth* burned areas.

**Results.** The experimental results are presented in Table 2, which shows that the model incorporating all variables slightly outperforms the baseline model relying only on the ignition as input. In Figure 3 we see two examples from the models' predictions in the test set. It is notable that models predict higher spread danger around the ignition point, with the model that uses input from all variables showing some enhanced skill compared to the no-skill baseline. To improve on the task, it might be important to include variables before and after the fire starts and not only from the date of ignition. Notably, Coffield et al. [66] find that the performance can be improved using weather data forecasts for $1 - 5$ days after the start of a fire. The dataset extraction script that accompanies the track includes the capabilities to extract datasets that include time series of arbitrary length before and after the start of each fire.

## 5    Potential other tracks to explore

Mesogeos incorporates burned areas and their sizes, as well as ignition points as predictands, and thus can be used for various ML applications. Thus, users can address additional tracks beyond those presented in the current study. The following ideas serve as suggestions for potential paths of research.

**Fire size prediction.** This application involves predicting the final size of fires rather than the presence of burned areas, which was addressed in this work. This can be framed as either a regression or a multiclass classification task. As a regression task, the target variable would be the burned area in hectares and therefore predictive models could be developed to estimate the final size of fires. As a classification task, a model could be employed to categorize fires into small, medium, or large. Notably, the dataset initially created for the wildfire danger forecasting track can be used as is for these two tasks.

**Extreme events forecasting.** Forecasting extreme events holds significant importance for effective fire management strategies. Concerning the Mediterranean, most of the damage caused by wildfires is the result of only a few large fires [67]. Therefore, anticipating these wildfires is crucial for ecosystem preservation and optimization of fire management. Mesogeos includes extreme events, such as the 5 massive fires in Greece in the summer of 2021, that burned nearly $94,000$ hectares [68], the extreme events in Portugal in October 2017, with 5 events burning more than $18,000$ hectares each [69] and the fire in Valencia, Spain in 2012 that burned over $50,000$ hectares [70]. Extracting a dedicated dataset for this track offers an opportunity to develop models explicitly predicting these extreme events. As these extreme events are rare, one potential approach to tackle this task would be treating it as an outlier detection problem.

**Wildfire susceptibility mapping.** Wildfire susceptibility is defined as the static probability of wildfires in a certain area, depending on the characteristics of the terrain and prevailing meteorological conditions [71]. This task can be framed as a binary classification task. In this context, the dataset extraction process would involve identifying positive samples as the number of all pixels affected by fire over multiple years. Negative samples would be obtained from pixels that have never experienced burning, i.e. not belonging to any burned area of the dataset. An ML model could then be employed to distinguish between the fire-susceptible and non-susceptible samples.

**Self-supervised learning.** The vast amount of data in the datacube remain untapped when extracting task-specific datasets. Self-supervised learning (SSL) [72] offers a promising approach to take advantage of the full capabilities of these data. SSL allows for acquiring a representation that can be utilized across various downstream tasks, including those mentioned earlier. Concerning the SSL track, extracting specific datasets is unnecessary, as the training samples can be directly extracted from the datacube during the data loading process. Careful engineering is essential when selecting and extracting samples to minimize the time required for the model to be trained.

**Modeling at different spatio-temporal scales.** Mesogeos has a resolution of $1km \times 1km \times daily$, enabling the examination of problems at that specific temporal and spatial scale. However, the flexibility of the datacube format allows for resampling in various temporal and spatial dimensions through appropriate aggregations. This would enable the treatment of other tasks in coarser temporal or spatial scales, like seasonal or sub-seasonal fire modeling.

**Beyond traditional ML.** When doing ML-based wildfire modeling for decision support it is many times important to dive deeper into understanding the underlying processes that drive the models' predictions. In that respect, Mesogeos can foster the development of explainable AI techniques [73] toward a better understanding of models and subsequently the interactions of the fire drivers that result in wildfires. Additionally, causal inference methods [74] could be used to assess the effects of human controls, such as agricultural practices or land use on wildfire regimes. Moreover, considering the stochastic nature of fire processes, noise commonly appears on the labels. Methods that take into account the noisy labels [75] and especially those estimating the inherent aleatoric uncertainty [76], could enhance the reliability of the models and support the decision-making. When existing layers in the datacube are not enough, the datacube can be easily enhanced with extra information such as socio-economic factors, settlement, and infrastructure.

## 6    Limitations

Despite the advantages of Mesogeos, we would like to acknowledge certain limitations. Firstly, the datasets inherit inaccuracies of the original data sources. Factors such as the satellites' spatial resolution and missing data resulting from cloud cover can influence the precise determination of the fires' location and size. Additionally, the acquisition of the fire ignition date is challenging and prone to deviations, as discussed in Section 3. Another limitation arises from the types of fires included in the burned areas' products. It is possible that there are fires resulting from prescribed burning or agricultural burning [41], which cannot be modeled using just the variables within Mesogeos. Moreover, while the daily temporal and $1km \times 1km$ spatial resolution of the datacube is appropriate for the applications suggested in this research, it cannot be used to address other, nowcasting-type problems related to other cycles of fire management such as fire spread, fire detection, or fire recovery related applications. For the target variable, Mesogeos considers the highly reliable final burned areas from EFFIS, but ignores intermediate temporal information of the wildfire evolution, compared to work explicitly targeting wildfire spread [20, 45, 44]. Furthermore, the dataset lacks information regarding fire suppression efforts. The interventions of firefighters and responders have the potential to influence fire dynamics both at the time of ignition, achieved through water application to weaken fire spread, as well as during winter months by means of fuel cleaning or controlled burning. It should be acknowledged that the absence of such data could influence the modeling of some of the tracks. For example, when training the model to predict fire danger, it remains unknown whether the fire would have grown larger (indicating higher danger) in the absence of any wildfire suppression measures, or the opposite. Finally, it is important to highlight that while ML can assist in wildfire modeling, an operational application in wildfire management necessitates a thorough evaluation across fire seasons and against operational baselines, including domain experts and wildfire responders in the process.

## 7    Availability and Maintenance

The Mesogeos datacube and the datasets utilized in this study are made publicly available. The project's website https://orion-ai-lab.github.io/mesogeos/ will hold updated links to the data and code repository, as well as a leaderboard for the ML tracks. The repository contains code for generating Mesogeos, extracting datasets for the tracks, and running the models, enabling the reproduction of the results presented in this work. We encourage the community to further contribute with more ML tracks and models and advance data-driven wildfire modeling using the Mesogeos datacube.

## 8    Conclusion

In conclusion, this work introduces Mesogeos, a valuable resource for data-driven wildfire modeling. By leveraging the structure of a datacube and incorporating variables that represent various fire drivers and historical wildfires, Mesogeos facilitates the extraction of diverse datasets, empowering researchers to model various fire-related tasks. In this work, we demonstrate two tracks focusing on fire danger forecasting and burned area prediction to showcase the effectiveness and potential of the dataset. Lastly, we present several alternative tracks that address a wide range of challenges and tasks associated with anticipating and understanding wildfires, thereby paving the way for new avenues of research and advancement in wildfire modeling.

## Acknowledgments and Disclosure of Funding

This work has received funding from the European Union's Horizon 2020 Research and Innovation Projects DeepCube and TREEADS, under Grant Agreement Numbers 101004188 and 101036926 respectively.

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
