# A    Supplementary Material

In the Supplementary Material we:

- Provide a detailed description of the dataset following the "datasheet for datasets" [1].

- Provide the details for the deep learning models that were used in the paper.

## A.1    Dataset Description

This subsection provides a detailed description of the dataset, providing the Motivation, Composition, Collection Process, Preprocessing, Uses, Distribution, and Maintenance, which are the key stages of the dataset lifecycle.

### A.1.1    Motivation

Mesogeos is created to enable the development of data-driven wildfire modeling in the Mediterranean. Wildfires are one of the most threatening natural phenomena and their impact is expected to aggravate even more due to climate change. While several physical models are used in various fire-related applications, they do not have the capabilities to model the complex interactions between the different fire drivers, like the data-driven models. However, the lack of comprehensive datasets hinders the widespread adoption of data-driven modeling in wildfire-related applications. Besides, even the existing datasets in the domain are restricted to a specific ML application. Thus, Mesogeos is introduced as a multi-purpose dataset that can be used for the development of several applications. Moreover, it covers the Mediterranean, one of the most fire-prone regions on Earth, where no other wildfire-related datasets are available.

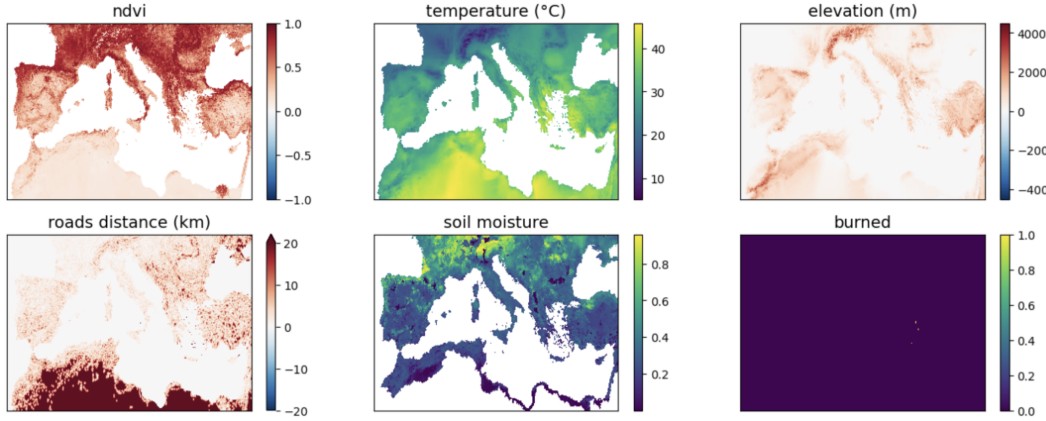

Figure 1: Visualization of some of the variables in Mesogeos datacube for a day.

Mesogeos was developed by the Orion Lab of the Institute for Astronomy, Astrophysics, Space Applications, and Remote Sensing of the National Observatory of Athens in collaboration with the Image Processing Laboratory (IPL) of the University of Valencia. This work has received funding from the European Union's Horizon 2020 Research and Innovation Projects DeepCube and TREEADS, under Grant Agreement Numbers 101004188 and 101036926 respectively.

### A.1.2    Composition

The Mesogeos dataset is structured in a spatio-temporal grid format, namely a datacube with three dimensions: longitude, latitude, and time. The datacube encompasses 27 variables related to known fire drivers such as meteorology, vegetation, land cover, and human activity. Mesogeos also includes historical burned areas, ignitions, and burned area sizes as separate variables. It has $1km \times 1km \times daily$ resolution and contains the values of the variables covering the period from 2006 to 2022. It

31 incorporates data from the wide Mediterranean area and spans a total area of $4714km \times 1753km$
32 and 6026 days, for a total of 47.796.706.692 different data points for each dynamic variable.

33 The detailed list of the variables with their input sources, original temporal and spatial resolution, and
34 their units are provided in Table 1.

35 We present a visualization of some of the variables in the datacube for a day in Figure 1.

Table 1: Table of all the variables in Mesogeos.

| Variable | Source | Sp. Res. | Temp. Res. | Units |
|---|---|---|---|---|
| *Dynamic variables* | | | | |
| Max Temperature | ERA5-Land | 9km | hourly | K |
| Max Wind Speed | ERA5-Land | 9km | hourly | m/s |
| Max Wind Direction | ERA5-Land | 9km | hourly | ° |
| Max Dewpoint Temperature | ERA5-Land | 9km | hourly | K |
| Max Surface Pressure | ERA5-Land | 9km | hourly | Pa |
| Min Relative Humidity | ERA5-Land | 9km | hourly | %/100 |
| Total Precipitation | ERA5-Land | 9km | hourly | m |
| Mean Surface Solar Radiation Downwards | ERA5-Land | 9km | hourly | $J/m^2$ |
| Day Land Surface Temperature | MODIS | 1km | daily | K |
| Night Land Surface Temperature | MODIS | 1km | daily | K |
| Normalized Difference Vegetation Index (NDVI) | MODIS | 500m | 16-days | - |
| Leaf Area Index (LAI) | MODIS | 500m | 8-days | - |
| Soil moisture | EDO | 5km | 10-days | - |
| Burned Areas | EFFIS | 1km | vector | $\{0, 1\}$ |
| Ignition Points | MODIS | 1km | vector | hectares |
| *Semi-static variables* | | | | |
| Population | Worldpop | 1km | yearly | $people/km^2$ |
| Fraction of agriculture | Copernicus CCS | 300m | yearly | %/100 |
| Fraction of forest | Copernicus CCS | 300m | yearly | %/100 |
| Fraction of grassland | Copernicus CCS | 300m | yearly | %/100 |
| Fraction of settlements | Copernicus CCS | 300m | yearly | %/100 |
| Fraction of shrubland | Copernicus CCS | 300m | yearly | %/100 |
| Fraction of sparse vegetation | Copernicus CCS | 300m | yearly | %/100 |
| Fraction of water bodies | Copernicus CCS | 300m | yearly | %/100 |
| Fraction of wetland | Copernicus CCS | 300m | yearly | %/100 |
| *Static variables* | | | | |
| Roads distance | Worldpop | 1km | static | km |
| Elevation | COP-DEM | 30m | static | m |
| Slope | COP-DEM | 30m | static | rad |
| Aspect | COP-DEM | 30m | static | ° |
| Curvature | COP-DEM | 30m | static | rad |

36 Mesogeos inherits inaccuracies of the original data sources. Many of the data come from satellites (e.g.
37 burned areas and fire ignitions, land surface temperature (LST), NDVI, LAI. Thus satellites' spatial
38 resolution and missing data from cloud cover cascade to the datacube. Moreover, due to satellites'
39 unavailability for specific days, there are some data missing for the LST MODIS product that was
40 replaced by the values of the previous days (172nd of 2006, 190th of 2015, 265th of 2021, 344th of
41 2019, 85th of 2017, 169th of 2019, the dates are in Julian date format). The same holds for the 233rd
42 day of 2019 for the LAI product. Moreover, the burned areas dataset provided by EFFIS does not
43 offer the same coverage of burned areas across all Mediterranean countries throughout the specified
44 timeframe, resulting in the absence of data for certain countries in specific years. The distribution
45 in the years from the EFFIS product is provided in Figure 2. However, EFFIS is continuously
46 post-processing fires from the countries spanning in the past, thus when they are made available, our

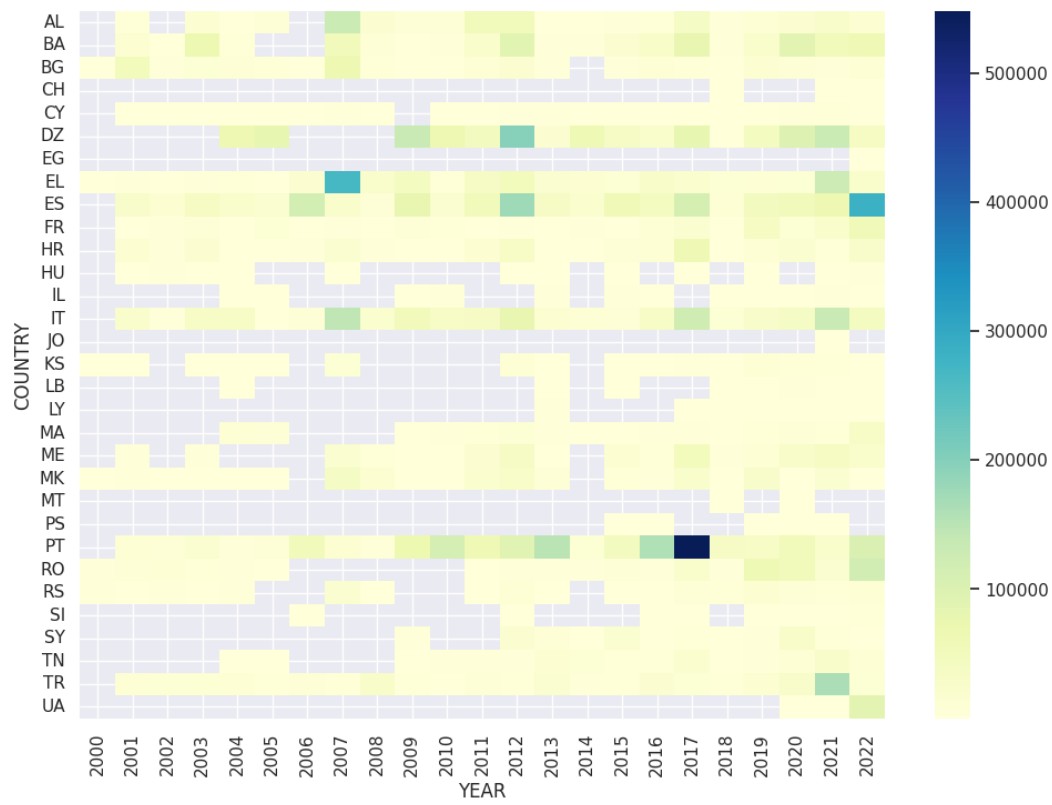

Figure 2: The distribution of the number of fires per country in the burned areas product provided by EFFIS.

intention is to append them in the datacube. In the meanwhile, if as more accurate products that refer to fires are produced, they can easily be appended to the datacube as separate variables.

As also noted in the main text according to the EFFIS website, the start dates of the fires provided by EFFIS may not always correspond to the date of ignition of the fire. For this, we make a specific post-processing to acquire the first detection of a wildfire, as this is very important in wildfire modeling. Despite the improvements achieved through our approach, it is essential to recognize that some misalignments in the ignition dates may persist in the datacube.

From the multi-purpose dataset Mesogeos, in this work, we extract two ML-ready datasets which are created by sampling the original datasets. One tailored to fire danger forecasting and one to final burned area prediction. The sampling scheme for the two datasets is analytically presented in the main paper. Moreover, the training, validation, test splits are also provided in the main text.

### A.1.3 Collection Process

All the data in Mesogeos were collected directly from the data sources in the format originally provided by each data source, and a recent crawling was used close to the final date of the datacube, September 29, 2022. The collection process for each of the data sources used for Mesogeos was different for each product:

- **ERA5-Land:** The hourly data from ERA5-Land were collected from https://cds.climate.copernicus.eu/cdsapp#!/dataset/reanalysis-era5-land?tab=form using the API request tool provided by the platform. Access to data is based on a principle of full, open, and free access (https://cds.climate.copernicus.eu/api/v2/terms/static/licence-to-

use-copernicus-products.pdf). The days and the timespan used was decided to match the shape of the datacube.

- **MODIS:** The data from MODIS (day's/night's land surface temperature, NDVI, LAI) were downloaded from NASA's portal https://modis.gsfc.nasa.gov/data/ using the official python script provided for downloading that needs to declare the days, tiles and products that need to be downloaded. There are no restrictions on the use of data (https://lpdaac.usgs.gov/data/data-citation-and-policies/). We downloaded the data covering the temporal and spatial dimensions of the datacube. Specifically, we downloaded tiles h17v04, h18v04, h19v04, h20v04, h17v05, h18v05, h19v05, and h20v05 for all the years from 2006 to 2022.

- **European Drought Observatory (EDO):** The soil moisture index was downloaded by a simple wget command from the EDO website https://edo.jrc.ec.europa.eu/gdo/php/index.php?id=2112. The data are distributed under a free open access (https://edo.jrc.ec.europa.eu/gdo/php/index.php?id=2044). We downloaded the whole data covering the timespan of the datacube.

- **European Forest Fire Information System (EFFIS):** The EFFIS data were downloaded from the real-time updated burnt areas database of the Data and Services tab of the EFFIS website https://effis.jrc.ec.europa.eu/applications/data-and-services. The data are distributed under the Creative Commons Attribution 4.0 International License. From all the data we used the data from 2006 til 2022, as these are the data with corrected dates. The previous years have not been processed yet from the EFFIS, as we noticed that all dates of all the fires were dated as 01/01/year.

- **Ignition Points:** The ignition points were downloaded from NASA's FIRMS download archive https://firms.modaps.eosdis.nasa.gov/download/create.php portal by a manual request for the specific region of interest and timespan. There are no restrictions on the use of data (https://lpdaac.usgs.gov/data/data-citation-and-policies/).

- **Land Cover:** The land cover data were downloaded from https://cds.climate.copernicus.eu/cdsapp#!/dataset/satellite-land-cover?tab=form for the years 2006-2022 using the API request tool provided by the platform. Access to data is based on a principle of full, open and free access https://cds.climate.copernicus.eu/api/v2/terms/static/licence-to-use-copernicus-products.pdf.

- **Worldpop data:** Roads distance and population data were downloaded via wget from https://www.worldpop.org for each country in the datacube's spatial span separately. Then the data from the separate countries were merged in a single grid using the GDAL (https://gdal.org/) merge command. The data are distributed under the Creative Commons Attribution 4.0 International License. The population data were provided yearly, thus the data that were downloaded were from 2006 to 2022, while for the road distance product, there is one single instance.

- **Digital Elevation Models:** The elevation product was downloaded from COP-DEM https://spacedata.copernicus.eu/en/web/guest/collections/copernicus-digital-elevation-model using a wget script. The product is available under a free license https://spacedata.copernicus.eu/collections/copernicus-digital-elevation-model. Then a post-process was used to generate slope, aspect, and curvature as described in the next section. This product is static.

It is unknown to the authors of the dataset if any ethical review processes were conducted by the providers of the datasets.

### A.1.4 Pre-processing, Cleaning, Labeling

We store the collected data in an internal machine and post-process them in order to append them in the 1 km × 1 km × 1 day resolution datacube.

- **ERA5-Land Data.** We combine 2 m dewpoint temperature (DT) and 2 m temperature (T), to calculate relative humidity using the following equation:

$$100 * \frac{\exp \frac{17.625*DT}{243.04+DT}}{\exp \frac{17.625*T}{243.04+T}} \tag{1}$$

  We also combine 10 m wind $u$-component and 10 m wind $v$-component to compute wind speed and direction. Then, we calculate the maximum daily value of 2 m temperature, 2 m dewpoint temperature, and surface pressure, the minimum daily value of relative humidity, the mean value of the surface solar radiation downwards, and the day's total precipitation. All these are calculated based on the hourly values of the day. For the wind, we calculate daily values of maximum wind speed. The daily wind speed direction is calculated at the hour that the maximum wind speed occurs. Finally, we use linear interpolation to map the 9 km spatial resolution to the 1 km spatial resolution of the datacube.

- **LAI, NDVI.** We concatenate the separate tiles downloaded from MODIS and we use the nearest interpolation to map these variables to 1 km spatial resolution. Moreover, as the time resolution of the products is 8 and 16 days, we forward-fill the values in time, to fill the temporal gaps and let the variables have a daily temporal resolution.

- **Day and Night Land Surface Temperature.** No pre-processing was conducted for these products.

- **Soil Moisture Index**. We reproject this variable in EPSG:4326 and we use the nearest interpolation to map it to 1 km spatial resolution. As the temporal resolution of the variables is 10 days, we also forward-fill it in time, to fill the temporal gaps.

- **Roads Distance** is a static variable that has no time dimension. No pre-processing is needed.

- **Yearly population density**. We collect each year's population density covering the years 2006-2020. As the years 2021, and 2022 are not available, we use the year's 2020 value as a proxy for them.

- **Elevation, Slope, Aspect, Curvature**. After gathering the elevation, we upscale it to a 1 km spatial resolution by using mean aggregation. After having the 1 km spatially resolved elevation at hand, we calculate slope, aspect, and curvature at 1 km spatial resolution, using GDAL https://gdal.org/.

- **Land Cover**. The land cover product comes in 300 m spatial resolution. In order to take a 1 km spatial resolution product, we create 8 variables, each one related to the fraction of one out of 8 classes of interest, which are based on the following subclasses: agriculture, forest, grassland, settlements, shrubland, sparse vegetation, water bodies, wetland. For this, we calculate the percentage of each subclass in the 1km pixel of interest. Thus, in each 1 km spatially resolved cell, each of these 8 variables has a value between 0 and 1, which represents the fraction of the class presence in the pixel. We repeat this process for all the years.

- **Burned Areas & Ignition Points.** The gathered shapefiles representing final burned areas, produced by different ignitions need some pre-processing to identify the date of the ignition of the fire and to combine several burned areas, that are produced by the same fire event, into one. To this end, we associate burned area products with the product of active fires. In order to identify the date of the ignition, we create a 1 km buffer around the burned area shapefiles and we look for an active fire inside this area that has been produced within a week's temporal buffer from the date that EFFIS provides for the burned area. Moreover, to combine several burned areas into one, we also use a buffer of 1 km and we do the following: We use as an anchor a burned area and we calculate its ignition point following the procedure above. If this ignition point is also lying inside buffered burned areas other than the anchor one and these burned areas have a date within a week before the anchor one, we consider that they belong to the same event as the anchor one. Having the ignition points and burned areas at hand, we rasterize them in the same 1 km spatial grid of all the input variables. Each ignition point also holds the burned area size of the fire.

165 The final datacube has 29 variables.

166 We shift backward the variables related to fire events (burned areas, ignition points). Moreover, we
167 also shift backward the meteorological data, as we assume that will be available through forecasts for
168 the day of interest.

### A.1.5  Uses

170 The dataset has already been used for two tasks: 1) next day's wildfire danger forecasting and 2)
171 final burned area prediction, given the fire ignition. The two tasks are analytically described in the
172 main text. The code for extracting the relative datasets and running the models is provided here:
173 https://github.com/Orion-AI-Lab/mesogeos. Moreover, alternative tracks for which the dataset can be
174 used are also presented in the main text. Some examples are final burned area size prediction, extreme
175 events forecasting, self-supervised learning techniques, wildfire susceptibility mapping, modeling on
176 different spatial and temporal scales, and causality or explainable AI methods.

### A.1.6  Distribution

178 The Mesogeos dataset is based on open datasets. We openly publish the Mesogeos datacube, the
179 derived datasets and models, and the code used to generate them at https://zenodo.org/record/7741518
180 under DOI: 10.5281/zenodo.7473331 and with a Creative Commons Attribution 4.0 International
181 License. Any work using the Mesogeos dataset should cite the main paper and the individual
182 dataset sources that have been used and listed in Table 1. We also provide a github reposi-
183 tory: https://github.com/Orion-AI-Lab/mesogeos and a website for the project: https://orion-ai-
184 lab.github.io/mesogeos/ with information on how to use the data and code.

### A.1.7  Maintenance

186 The dataset will be maintained by Spyros Kondylatos and Ioannis Prapas. If needed, the emails for
187 contacting are skondylatos@noa.gr, and iprapas@noa.gr. The dataset may be updated if needed (with
188 the inclusion of more years when the data are available, more layers for more applications, or more
189 fire data sets). The updates will be ad-hoc and will not be periodical. Each time the dataset will be
190 updated, there will be a new GitHub code release and a new version in Zenodo. The older versions of
191 the datacube will not be maintained. For contributing with new ML-ready datasets there are analytical
192 instructions in the GitHub repository https://github.com/Orion-AI-Lab/mesogeos in the Contributing
193 section.

### A.1.8  Author statement

195 The authors of this scientific paper bear full responsibility for any violation of rights that may arise
196 from the collection of the data included in this research.

### A.2  Models Details

### A.2.1  Track A: Next Day's Wildfire Danger Forecasting

199 Long Short-term Memory (LSTM), Transformer, and Gated Transformer are the models used for
200 Track A. All the hyperparameters included in this section were defined using the best-performing
201 model in the validation set. All the models were trained for 30 epochs with the binary cross-entropy
202 loss with $\ell_2$-norm regularization, the Adam optimizer, and a batch size of 256. The data were
203 normalized before serving as input. We filled with the temporal aggregate of the time series the nulls
204 existing in the inputs of the models. When the value was still null, we were filling it with 0, which is
205 the mean of each variable, after being normalized.

206 Regarding LSTM's architecture, a normalization layer is followed by an LSTM layer with 128
207 neurons, which is followed by two linear hidden layers with 128 and 64 neurons, respectively, and
208 an output 2-class softmax layer. The final model is trained with a 0.0063 weight decay and 0.004

learning rate, which is divided by 10 every 15 epochs. All linear layers, but the last, are followed by a dropout with probability $p = 0.5$ and the ReLU activation function.

Regarding Transformer's architecture, the time-series input is passed through a linear layer with 256 neurons and then through a standard positional encoding layer. Then, it is followed by 2 standard Transformer Encoder Layers which are made up of self-attention and feed-forward layers. The number of heads in each layer is 2 and the neurons in each feed-forward layer are 512. The final layer is a 2-class softmax layer. The final model is trained with a 0.0018 weight decay and 0.00029 learning rate, which is divided by 10 every 15 epochs.

Regarding Gated Transformer's architecture, for the time-Transformer block, the original time-series input is passed through a linear layer with 256 neurons and then through a standard positional encoding layer. This is followed by 4 standard Transformer Encoder Layers which are made up of self-attention and feed-forward layers. The number of heads in each layer is 4 and the neurons in each feed-forward layer are 512. Regarding the channel-Transformer block, following the original paper, the time-series input is transposed in order for the attention to act in the dimension of the variables. The input is then passed through 4 Transformer Encoder Layers with the same parameters as the time one. The outputs of the two Transformers blocks are then concatenated and passed through a softmax layer. The first logit of the softmax is multiplied with the outputs of time-Transformer and the second by the output of the channel-Transformer. The final outputs are concatenated and pass through the final 2-class softmax layer. The final model is trained with a 0.0045 weight decay and 0.00012 learning rate, which is divided by 10 every 15 epochs.

All the models were trained in a system with 2 GPUs (NVIDIA GeForce RTX 3080), each one having a memory of 10 GB. The total RAM of the system is 128GB and it also has 48 CPUs. The models were trained using one of the GPUs.

All models were developed using the PyTorch library [2]

### A.2.2 Track B: Final Burned Area Prediction

For track B, the U-Net was employed.

A standard U-Net model was used with an EfficientNet-B1 Encoder, using Pytorch [2], Pytorch Ligthning [3] and the segmentations_model_pytorch library [4]. The model was trained for a maximum of 50 epochs, using early stopping, with the binary cross-entropy loss with $\ell_2$-norm regularization, the Adam optimizer, and a batch size of 128. The standard U-Net architecture was used with no modifications. The final model is trained with a 0.000001 weight decay and 0.005 learning rate. The data are scaled with the minimum and maximum values in the range [0, 1] before being served as inputs to the model. We fill the nulls with -1. The model was trained in a system with 2 GPUs (NVIDIA GeForce RTX 3090), each one having a memory of 24 GB. The total RAM of the system is 128GB and it also has 32 CPUs. The models were trained using one of the GPUs.