# OpenReview forum: "Mesogeos: A multi-purpose dataset for data-driven wildfire modeling in the Mediterranean"
_NeurIPS.cc/2023/Track/Datasets_and_Benchmarks — NeurIPS 2023 Datasets and Benchmarks Oral_

### Official Review · Reviewer_cj7g · 2023-07-21
**Important wildfire dataset but several clarifications are needed**

**Rating:** 7
**Confidence:** 5
**Clarity:** The paper is well-written.

**Strengths:**

The dataset is comprehensive, large, and shows several important demonstrations of how it can be used (unlike prior work). The area under coverage is also novel, as other large fire spread databases (e.g., WildfireDB) have mostly focused on North America.

**Additional Feedback:**

N/A.

**Correctness:**

The claims are mostly correct, other than some clarifying questions that I pointed out above.

**Documentation:**

Yes.

**Limitations:**

See above.

**Opportunities For Improvement:**

I have the following questions/suggestions:

1. I am concerned about how prior work is presented; e.g., references 36 and 37 are said to have used the "inconsistent" MOD14A1 product as the target variable. VIIRS has a higher resolution than MODIS (375 vs. 1000m per pixel) and increased night-time fire detection performance. While MODIS does have its advantages (e.g., crisp background detection that can translate to improved fire detection), this distinction and the advantages and disadvantages of prior work should be clearly highlighted. The claim that VIIRS is inconsistent compared to MODIS (that the current paper uses) is unsubstantiated, especially when a detailed discussion is omitted.

2. On a related note, how does the proposed dataset differ from references 36 and 37 other than the source from which the target variable was obtained?

3. Fire occurrence and spread data are typically extremely imbalanced, which in turn results in poor performance of off-the-shelf predictive models. The data mentioned in this paper is surprisingly balanced, e.g., the final dataset has 33.3% positives as opposed to less than 5% positives in prior wildfire databases (e.g., WildfireDB). Can the authors clarify why this is the case?

4. Two important bottlenecks in modeling data-driven models for fire spread are:
a) Modeling Response: A lot of wildfires are actually stopped from spreading by firefighters, potentially confounding the data about intrinsic fire spread conditional on environmental parameters such as weather, land patterns, etc. How do the authors include this? I acknowledge that collecting data about interventions is hard, but I would encourage the authors to mention this and potentially include a discussion.
b) Modeling new fires vs spread: The distinction between whether a fire spread to a neighboring spatial location versus a new fire started in hard to model. In predicting spread over large areas such as 64km x 64km, how do the authors take this challenge into consideration?

**Relation To Prior Work:**

Yes, other than clarifying questions mentioned above.

**Summary And Contributions:**

The dataset potentially benefits the machine learning, climate science, policy, and fire response communities. However, there are several clarifications needed. I am open to having a discussion in the author response period to change my score/understand the paper better.

---

> ### Author Response · Authors · 2023-08-14
>
> > The claim that VIIRS is inconsistent compared to MODIS…is unsubstantiated.
>
> To the best of our knowledge, we have not made any claims in the paper that VIIRS is inconsistent compared to MODIS. However, we believe this comment arose due to the fact that we had wrongly clustered works 43, 44, 45 stating they all use MOD14A1. Only Huot et al uses that product, while the two other use VIIRS. We have corrected that part (lines 104-106), and we thank the reviewer for raising our attention to this. Additionally, the comment allowed us to reflect on the fact that it is important to discuss the VIIRS and MODIS active fire and burned areas products that most wildfire datasets depend on. This has been expanded in the related work section (lines 68-88).
>
> > 1) I am concerned about how prior work is presented…
> > 2) How does the proposed dataset differ from references 36 and 37 other than the source from which the target variable was obtained?
>
> We have made significant revisions to the Related Work section. We appreciate the reviewer’s suggestion, as we believe it allowed us to improve the clarity and comprehensiveness of this section.
>
> For 1), we incorporated information about the VIIRS satellite along with its advantages compared to MODIS and we introduced a discussion about the VIIRS and MODIS products in lines 68-88, declaring also their limitations.
>
> For 2), we introduced a paragraph, comparing our dataset with the other existing datasets, including datasets 36-37. Specifically for the distinction of our datasets we note (see also lines 110-125):
>
> 1) Mesogeos is a multi-purpose dataset that can be used to address several tasks, while the two datasets 36-37 are tailored for predicting wildfire spread. This task cannot be treated with Mesogeos because it contains accurate information on the burned area but no information on the evolution of the wildfire (see lines 355-360).
> 2) Mesogeos incorporates a wide range of daily inputs covering fire drivers across the entire region of interest, while 36-37 introduce data points related only to specific fire events.
> 3) In Mesogeos, we use the burned areas product from EFFIS as an anchor for a fire event, refining it using the MODIS active fires, while datasets 36 and 37 use an active fire product as the target of the modeling
> 4) Mesogeos covers the Mediterranean, while 36-37 the USA
>
> Specifically for point 3, we would like to provide some more clarifications, as this is also related to question 4. In datasets 36, 37 the active fire product is used as the target. This induces uncertainty regarding when a hotspot corresponds to a specific fire event and when not (as highlighted in the Limitation section in 37). In Mesogeos, a burned area product (EFFIS) is used as a representative for each fire event. Then the MODIS active fire product is used only for the refinement of the date and the declaration of the ignition point (see lines 191-203). Although this choice of target may have limitations in addressing tasks like fire spread prediction as there is a lack of temporal information of the fire evolution (see lines 355-360), it ensures a more reliable target, which is derived from a post-processed burned area product.
>
> > 3) …The data mentioned in this paper is surprisingly balanced
>
> We have included a complete answer in the global response.
>
> > 4) collecting data about interventions is hard, but I would encourage the authors to mention this and potentially include a discussion…The distinction between whether a fire spread to a neighboring spatial location versus a new fire started in hard to model…how do the authors take this challenge into consideration?
>
> In response to the first part regarding the actions of firefighters, we would like to highlight that we have already included a small discussion addressing this matter in the Limitations section, specifically in lines 364-367. Following the reviewer’s comment we have further expanded this discussion in lines 361-364. We note, though, that firefighting practices vary across regions and evolve over time, as well as the fact that these data often remain proprietary and inaccessible to the public. Nonetheless, we would like to emphasize that if access to such data is available, they can be easily integrated in the Mesogeos datacube. This is facilitated by the fact that i) the repo contains code to expand the datacube and ii) the datacube itself includes precise metadata such as geographical information for each grid cell.
>
> For the second part, related to the distinction between fire spread and new fire occurrence, firstly we should highlight that the anchor for defining a fire event in this dataset is the final burned area from EFFIS and not the active fire product. As such, the dataset does not contain any intermediate information about the spread, and each wildfire event is represented by its final burned area (this has been explicitly added in the limitations in lines 358-360, see answers 1, 2 above).

---

> > ### Comment · Reviewer_cj7g · 2023-08-24
> > **Reviewer Comment**
> >
> > I am happy with the rebuttal and have reflected it in my score. I am still a bit unclear about the data distribution. Can the authors explain why the data is significantly more balanced than standard wildfire datasets?

---

> > > ### Author Response · Authors · 2023-08-24
> > >
> > > We are glad that the reviewer has updated the score after our response.
> > >
> > > > Can the authors explain why the data is significantly more balanced than standard wildfire datasets?
> > >
> > > This has been covered in the global response addressed to all reviewers. In a nutshell:
> > > - The Mesogeos cube is not balanced, covering the whole Mediterranean. Fire is a "rare" phenomenon spatiotemporally and this is evident in the Mesogeos cube.
> > > - Dataset for Track A is sampled from the Mesogeos cube and is indeed balanced as a choice, similar to related literature (citations 1, 2, 3, 4 in the global response).
> > > - This split is intended to be used for benchmarking the performance of different ML models and not to demonstrate the real-world performance
> > >
> > > Moreover, we would like to highlight that in Track A we sample non-burned cells (aka negatives) alongside fire events (positives) for the classification task, unlike standard wildfire spread literature (43, 44, 45) where datasets typically only consist of fire events and the imbalance occurs from the dominance of small fires.

---

> > > > ### Comment · Reviewer_cj7g · 2023-08-28
> > > > **Reviewer Comment**
> > > >
> > > > Thank you for the response. This makes sense to me.

---

> > ### Comment · Reviewer_vRfc · 2023-08-28
> >
> > I would like to thank for the authors for their responses in the rebuttal and addressing the points raised in my review. I leave my rating for the paper the same.

---

### Official Review · Reviewer_vRfc · 2023-07-23
**A well written and timely paper introducing a wildfire modelling dataset**

**Rating:** 7
**Confidence:** 4
**Clarity:** The paper is well-written and very ea…

**Strengths:**

The strengths of the paper are:

- The nicely presented overview of the application domain and the well written nature of the paper.
- The introduction of two standard but challenging tasks to the ML community within the wildfire modelling community.
- The datacube is a comprehensive spatio-temporal set of inputs to base wildfire modelling on.

**Additional Feedback:**

see rest of the comment.

**Correctness:**

- The training and modelling of the task used in Task 1 is a somewhat confusing:

       In line 193  the authors describe how they weight the CE loss relative to burned area size to "learn to assign greater danger to larger fires". This is somewhat confusing as it is stated that Task 1 is treated as a binary classification problem for each cell in the grid:  danger of fire/ no fire. Normally for a weighted CE loss a separate weight is assigned to each class typically to cope with the class imbalance problem. However, instead in the paper each training example is given its own weight. How exactly is this weighting used? Is some form of importance sampling performed so that grid cells associated with larger fires more frequently sampled during training? Some clarification on this point would be great. Also does the weighting imposed make it harder to detect small fires and is this the desired outcome?

      It would appear that formulating the problem as a more fine-grained multi-label (encoding level-of-danger and/or potential area of spread of fire) or as a regression problem would make for a set-up more in keeping with the desired outcome of the authors.

A few minor points:

- Task 2 is considered as a segmentation task. It would be good if the metrics reported in table 2 also included some of the  more common metrics associated with semantic segmentation such as F1 score, IoU etc... Were other baseline architectures considered for this task apart from the U-Net?

- In Task 1 the input to each classifier is a temporal sequence of features from one spatial grid cell. Would considering spatially neighbouring inputs be beneficial or would it make the training harder?

- Also in task 1 at test time the negative samples are somewhat sparsely sampled and there are only twice as many negative samples as positive samples. I would presume in most real world situations the number of negative samples would be far more than positive ones.  How much harder would this increase in the imbalance between positive and negative samples at test time make the problem and would the false positive rate increase a lot?

- Lines 157-160: How much data is missing where?

**Documentation:**

Yes.

**Ethics:**

No.

**Limitations:**

- There is probably noise in the labels both in the training, validation and test splits due to the factors explained in the paper. However, how, hard would it be to "fix" and or clean-up at least the labels for the test sets?

**Opportunities For Improvement:**

- The dataset is huge w.r.t. memory usage. It would be nice to have a smaller but still meaningful version of the dataset to allow it to be used be a broader set of researchers to use locally.

- The training for the predictor in Task 1 is a little confusing. It would be great if this could be made clearer. Also at the moment it is formulated as a binary classification problem, however, it appears the authors consider it as a more fine-grained classification problem that is the "level of danger". Would it make sense to more explicitly model the level of risk as a more multi-label classification problem with each class representing a danger level?

- The paper mentions that the MODIS AF product is not so accurate, but then it is still used to find the ignition date (in conjunction with the EFFIF data). Did the authors consider using the VIIRS AF instead or is the difference in resolutions make this not feasible?

**Relation To Prior Work:**

Yes this is well described.

**Summary And Contributions:**

This paper introduces the large-scale spatio-temporal Mesogeos Datacube (1km x 1km x 1day resolution). It consists of weather measurements, vegetation and human activity data from 2006-2022 in the Mediterranean region.  EFFIS and the MODIS active fire product are used to estimate when and where wildfires ignited and the final burnt area created by the wildfire. The wildfire modelling tasks involve the prediction tasks of  "wildfire danger forecasting" and "final burned area prediction" using input from the datacube.

The contributions of the paper are:
 - It introduces a "clean, standardized" spatio-temporal datacube with relevant data for wildfire prediction tasks. This will save the ML researcher the hassle of interacting with the tangled web of primary sources if they want to work on this important problem.
 - It highlights wildfire modelling tasks to the ML community with its well written introduction and related work sections, and the "Potential other tracks to explore" section. In tandem it sets researchers up with a manageable data-structure to experiment with. This is a critically important field as can be seen from recent and current wildfire events and ML conferences should be actively promoting work in this area.
 - The paper defines two standard wildfire tasks and provides reasonable baselines for each.

---

> ### Author Response · Authors · 2023-08-14
>
> > The dataset is huge w.r.t. memory usage.
>
> Though the dataset is huge, we facilitate its use:
> - To prevent the need for local downloading, we uploaded the dataset to a public S3 bucket for cloud-access and added instructions in GitHub on how to access it
> - The datacube is shared in a cloud-optimized zarr format that allows users to lazily access parts of it without loading it in memory
> - We provided the 2 ML datasets used for the 2 tracks, which are manageable in size.
> - We shared the code to extract the datasets.
>
> > more explicitly model the level of risk as a more multi-label classification problem?
>
> Wildfire danger has been many times tackled as a binary classification task (citations 20, 42, 60). Here, we also incorporate the information from the burned area size (lines 208-218, 230-236), to associate larger fires with higher danger. However, alternative formulations could be explored to model wildfire danger. These include a multi-class approach, where the labels could be extracted based on the size of the fires (lines 300-306).
>
> > consider using the VIIRS AF instead or is the difference in resolutions make this not feasible?
>
> We do not use the MODIS AF product as a proxy for fire ignitions. Instead, we determine the burned areas from the EFFIS as "correct", more reliable products. We then use the MODIS AF ONLY to find the ignition date and point of the fire.
>
> We did not incorporate VIIRS because:
> - VIIRS started in 2012, whereas the datacube spans from 2006 onwards. This gap would introduce an inconsistency in dataset collection
> - Mesogeos contains satellite variables (NDVI, LST) from MODIS. We want to avoid a wrong ignition date that may leak target data into these variables, aka an ignition date after the actual start of the fire. We ensure that using the MODIS AF to move the ignition date back, in case AF have been detected before the EFFIS-derived start of the fire.
>
> We expanded the Related Work in lines 68-88 and added a few lines in Section 3 (195-196, 200-202) for this.
>
> > How hard would it be to "fix" and or clean-up at least the labels for the test sets?
>
> In the context of burned areas, we have minimized the noise, using what we consider the best available product in Europe (EFFIS) and by refining it using the MODIS AF product.
>
> From an ML perspective, noise is mostly evident in the labels in the negatives. A potential approach for this would be to treat the task as a Noisy Labels problem (citation 75). The noise can also be associated with aleatoric uncertainty (citation 76). We added some lines for this (341 - 344) .
>
> > The training for the predictor in Task 1 is a little confusing
>
> We weigh the samples’ contribution to the loss according to the final burned area size, assuming that the larger fires are associated with higher danger. This emulates what is done in other works that is closer to importance sampling (citation 42) that uses all the grid cells affected by fire, implicitly weighing fire events by the number of cell grids burned. We use one representative grid cell and weigh it as follows: We multiply its loss with the burned area size that corresponds to the sample. The loss is getting higher, if it corresponds to a larger fire. In practice, in order to prevent the larger fires from dominating the learning process, we multiply with the log of the burned area size.
>
> This is better explained in lines 232-235.
>
> > ..semantic segmentation metrics... Were other baseline architectures considered for this task…?
>
> The objective is to estimate the likelihood of neighboring pixels being affected by a fire, following an ignition. Instead of a binary mask, it’s more interesting to differentiate between the more and less probable cells for the fire to expand to. Thus, traditional segmentation metrics were not used. Instead, we used metrics that can capture the skill of this differentiation ability. As this was not clear, we made some revisions (lines 260-261, 263-265, 267-268).
>
> We have not tried other baselines, as we found it out of scope for this paper to find the best models for the task, but rather a reasonable baseline. However, we encourage other practitioners to develop models that surpass it.
>
> > Would considering spatially neighbouring inputs be beneficial or would it make the training harder?
>
> We explored this in our prior work (citation 42) and found that spatial context did not significantly enhance the results. Thus, we decided to make it easier for users to work with a smaller dataset rather than creating a much larger dataset that may provide little or no gain. Practitioners can extract such a dataset, using the template extraction code.
>
> > How much harder would this increase in the imbalance between positive and negative samples at test time make the problem…?
>
> We have included an answer in the global response.
>
> > How much data is missing where?
>
> We present in the Supplementary Material (Figure 2) the distribution of the number of fires per country in EFFIS.

---

### Official Review · Reviewer_28Ft · 2023-07-27
**Review of Mesogeos**

**Rating:** 9
**Confidence:** 4
**Correctness:** Yes, that seems very good.
**Clarity:** The paper is very well and clearly wr…

**Strengths:**

See also above. The paper makes a highly valuable, and - sadly - timely/topical contribution given the current wildfires in the Mediterranean. Studying and forecasting wildfires will become an ever more important task, also to effectively address climate change adapation, for example.

I rarely have as few corrections for any paper. The authors wrote an excellent paper, making the big picture accessible while also reporting important details, e.g. on interpolation methods in the Supplementary. It was really a pleasure to read.

**Additional Feedback:**

N/A

**Documentation:**

From initial checks, this all seems fine.

**Ethics:**

The are no concerns from my side.

**Limitations:**

The limitations have been well discussed and addressed by the authors, well beyond what I would consider normal.

**Opportunities For Improvement:**

My only concern is the labelling of the colorbars in Figures 2 and 3. While labels low->high would probably suffice in TV broadcasting, it is not clear if this is a linear colorbar varying simply between values 0 and 1...number labels should be added.

Even more so, I am entirely left in the dark concerning the colours in Figure 3, because there is no colorbar, which should be added for scientific rigour.

**Relation To Prior Work:**

This seems appropriate to me.

**Summary And Contributions:**

The paper by Kondylatos et al. introduces a new dataset for data-driven wildfire modelling in the Mediterranean and compares several benchmark performances for two prediction tasks: (a) wildfire danger/risk forecasting and (b) burned area estimation following ignition.

To the best of my knowledge this dataset in its composition is highly novel, was further enhanced by the authors addressing a few source data limitations (e.g. start dates of fires), and will find many users across multidisciplinary communities.

---

> ### Author Response · Authors · 2023-08-14
>
> We are very glad for the positive feedback and agree with the reviewer’s suggestions that we implemented.
>
> Changes made: We have added colorbars in Figures 2 and 3 as suggested.

---

### Official Review · Reviewer_v8vR · 2023-08-24
**Mesogeos: A multi-purpose dataset for data-driven wildfire modeling in the Mediterranean**

**Rating:** 9
**Confidence:** 3
**Correctness:** This is a well written and constructe…
**Clarity:** Excellent writing, kudos to the authors!

**Strengths:**

Creation of the Mesogeos dataset and accompanying ML models.

**Additional Feedback:**

N/A

**Documentation:**

Documentation is sufficient.

**Ethics:**

I have no ethical concerns with the study.

**Limitations:**

I thought that the authors did a very good job describing the limitations.

**Opportunities For Improvement:**

Incorporation of more datatsets, which currently heavily relies on ERA5 and MODIS.

**Relation To Prior Work:**

Very nice Introduction and contextualization with prior work.

**Summary And Contributions:**

For the application of ML models collecting and formatting large and disparate datasets is critical but challenging.  The authors have done much of the heavy lifting by creating the Mesogeos dataset for wildfire prediction.  This is a societal beneficial enterprise on a very important hazard, wildffires in the Mediterranean region as exemplified by this summer.

---

### Author Response · Authors · 2023-08-14
**Global Response**

We would like to thank all the reviewers for their constructive feedback and suggestions, which have contributed to the enhancement of the quality of the manuscript.

Based on the comments, we have updated the manuscript PDF for revision. Additionally, we have included a version of the manuscript with the changes based on the reviewers’ comments inside the Supplementary Material zip file (the name of the file is revised.pdf). The changes are colored in red.

In this general response, we present the most important changes that have been implemented after the reviewers’ comments:

- The related work has been expanded with a better description of open satellite-derived data for burned areas and active fires, as well as a better comparison with previous work, highlighting distinctions between our dataset and existing ones.
- We have refined the Task Formulation and Experimental Setup paragraphs of the two ML tracks to offer a better explanation of the training processes. In Track A, we have also provided a more comprehensive explanation of the weighting scheme employed.
- We have changed/added the color bars in Figures 2 and 3
- Clarifying sentences have been added to Ignition Date Calculation paragraph of section 3 to better explain how the cross-reference between the EFFIS product and MODIS AF happens.
- We have made a slight modification to the Beyond traditional ML paragraph of the Potential Other Tracks to Explore section, to incorporate an approach focusing on Noisy Labels/Uncertainty estimation.
- We have expanded on the limitations of our study related to the lack of wildfire behavior data and information on fire suppression.

Finally, we would like to address a concern raised by both reviewers vRfc and cj7g regarding the dataset distribution in Track A. The concern of the two reviewers is that in real-world scenarios the data will be much more imbalanced than the 2:1 ratio between negatives and positives that are used in the current study. While we acknowledge that the imbalance is much higher in the real world than in the sampled datasets, our negative/positives split is well-suited to benchmark the performance of different ML models. This is in line with existing wildfire danger literature [1, 2, 3, 4]. Moreover, It is valid to say that the test set may not fully reflect real-world precision, but it does for recall. Using the Mesogeos cube, it is entirely possible to measure real-world precision (e.g. by measuring the performance of the entire region for some days). Furthermore, it is possible to calibrate any models developed with the sampled datasets [5]. Finally, we would like to stress that it is out of the scope of this work to provide operational solutions. To apply ML-based models in an operational context, one would need to measure real-world performance across fire seasons and extensively compare against non-ML baselines. (See lines 369 - 370). In that setup, the Mesogeos dataset can be exploited to investigate the use of ML methods for operational firefighting planning and decision-making.

[1] Bjånes, Alexandra, Rodrigo De La Fuente, and Pablo Mena. “A Deep Learning Ensemble Model for Wildfire Susceptibility Mapping.” Ecological Informatics 65 (August 1, 2021): 101397. https://doi.org/10.1016/j.ecoinf.2021.101397.

[2] Le, Hung Van, Duc Anh Hoang, Chuyen Trung Tran, Phi Quoc Nguyen, Van Hai Thi Tran, Nhat Duc Hoang, Mahdis Amiri, et al. “A New Approach of Deep Neural Computing for Spatial Prediction of Wildfire Danger at Tropical Climate Areas.” Ecological Informatics 63 (April 1, 2021): 101300. https://doi.org/10.1016/j.ecoinf.2021.101300.

[3] Zhang, Guoli, Ming Wang, and Kai Liu. “Forest Fire Susceptibility Modeling Using a Convolutional Neural Network for Yunnan Province of China.” International Journal of Disaster Risk Science 10, no. 3 (September 2019): 386–403. https://doi.org/10.1007/s13753-019-00233-1.

[4] Kondylatos, S., Prapas, I., Ronco, M., Papoutsis, I., Camps‐Valls, G., Piles, M., ... & Carvalhais, N. (2022). Wildfire danger prediction and understanding with Deep Learning. Geophysical Research Letters, 49(17), e2022GL099368.

[5] Pozzolo, Andrea Dal, Olivier Caelen, Reid A. Johnson, and Gianluca Bontempi. “Calibrating Probability with Undersampling for Unbalanced Classification.” In 2015 IEEE Symposium Series on Computational Intelligence, 159–66. Cape Town, South Africa: IEEE, 2015. https://doi.org/10.1109/SSCI.2015.33.

---

### Decision · Program_Chairs · 2023-09-22

**Decision:**

Accept (Oral)

**Comment:**

This submission introduces a novel dataset for wildfire modelling in the Mediterranean.

It consists of historical records for over 17 years of wildfire ignitions and burned areas. It further provides benchmarks for two tasks using the dataset: (i) wildfire danger forecasting and (ii) burned area estimation after ignition. Given the breadth and depth of the dataset and benchmark, the highly important topic, solid baselines, and the reviewers' feedback and excellent ratings, the decision is to recommend this paper for acceptance as oral presentation.